# Universal Health Coverage and the Pacific Islands: An Overview of Senior Leaders’ Discussions, Challenges, Priorities and Solutions, 2015–2020

**DOI:** 10.3390/ijerph19074108

**Published:** 2022-03-30

**Authors:** Adam T. Craig, Kristen Beek, Katherine Gilbert, Taniela Sunia Soakai, Siaw-Teng Liaw, John J. Hall

**Affiliations:** 1School of Population Health, University of New South Wales, Sydney, NSW 2052, Australia; k.beek@unsw.edu.au (K.B.); siaw@unsw.edu.au (S.-T.L.); john.hall@unsw.edu.au (J.J.H.); 2Nossal Institute for Global Health, School of Population and Global Health, University of Melbourne, Melbourne, VIC 3010, Australia; katherine.gilbert@unimelb.edu.au; 3Public Health Division, Pacific Community, Suva, Fiji; sunias@spc.int

**Keywords:** universal health coverage, Pacific Islands, primary health care, health systems, sustainable development goals, global health

## Abstract

In 1995, Pacific Health Ministers articulated their vision of a healthy Pacific as ‘a place where children are nurtured in body and mind; environments invite learning and leisure; people work and age with dignity; where ecological balance is a source of pride; and where the ocean is protected.’ Central to this vision is the achievement of universal health coverage (UHC). To provide an indication of the UHC-related priorities of Pacific health authorities and promote alignment of domestic and international investments in health sector development, we thematically analyzed the discussion, resolutions, and recommendations from 5 years (2015–2020) of senior-level Pacific health meetings. Five main themes emerged: (i) the Healthy Islands vision has (and continues to have) a unifying influence on action for UHC; (ii) adoption of appropriate service delivery models that support integrated primary health care at the community level are needed; (iii) human resources for health are critical if efforts to achieve UHC are to be successful; (iv) access to reliable health information is core to health sector improvement; and (v) while not a panacea for all challenges, digital health offers many opportunities. Small and isolated populations, chronic workforce limitations, weak governance arrangements, ageing and inadequate health facilities, and supply chain and logistics difficulties (among other issues) interact to challenge primary health care delivery across the Pacific Islands. We found evidence that the Healthy Islands vision is a tool that garners support for UHC; however, to realize the vision, a realistic understanding of needed political, human resource, and economic investments is required. The significant disruptive effect of COVID-19 and the uncertainty it brings for implementation of the medium- to long-term health development agenda raises concern that progress may stagnate or retreat.

## 1. Introduction

The 22 Pacific Island Countries and Territories (PICTs) have a population of 11.4 million people living on many hundreds of islands and atolls spread across one-third of the earth’s surface [1,2]. The PICTs vary considerably in population size and density, topography, development status, disease burden, and resource availability for health [3,4]. In 1995, at the 11th meeting of Pacific Health Ministers, health leaders articulated their vision of a healthy Pacific as ‘a place where children are nurtured in body and mind; environments invite learning and leisure; people work and age with dignity; where ecological balance is a source of pride; and where the ocean is protected’ [5]. The Healthy Islands vision has been hailed as ‘a truly ecological model of health promotion’ [6,7] that, after being reaffirmed twice (at 2015 [5] and 2017 [8] meetings of Pacific Health Ministers), remains a seminal guide for social, environmental, and health development in the Pacific today.

Central to the Healthy Islands vision is improving the health of Pacific populations, driven, in part, by an effort to advance universal health coverage (UHC) [9,10]. UHC is the concept that all people, regardless of who they are or where they live, have access to the full range of health services they need, when and where they need them without financial hardship [11]. Achievement of UHC is at the core of attaining the goal of ‘health for all’ set out in the Alma-Ata declaration of 1978 [12], and the more contemporary Sustainable Development Goals (SDGs) (specifically target 3.8) by 2030 [13]. The status of UHC in the Pacific, as indicated by the mean PICT Global Health Observatory ‘UHC index of essential service coverage’ score in 2017 was 50.4 (SD: 8.7) out of a possible 100, well below the mean level of other countries in WHO’s Western Pacific region (74.7 (SD: 12.3)) and of the world (64.0 (SD:15.5) [14]. The emergence of the coronavirus pandemic in 2019/20 and its associated impact has had severe and likely long-term ramifications of UHC implementation and priorities in the PICTs.

With the Healthy Islands vision, the SDGs, and UHC core concepts around which health system development efforts are framed in the Pacific, and dialogue about achieving them has been a feature of high-level regional public health meetings for many years. This paper analyzes the discussions, resolutions, and recommendations of regional health leader fora and the peer-reviewed literature to provide an indication of the UHC-related priorities of PICTs with the aim of promoting alignment of domestic and international investments in health sector development.

## 2. Materials and Methods

This desktop study had two parts.

The first part of the study involved the collection of 5 years (2015–2020) of publicly available final reports from Pacific Heads of Health meetings, Pacific Health Ministers meetings, and WHO Regional Committee meetings and extraction of data related to the UHC challenges, responses, and priorities of PICTs. A 5-year duration was considered sufficient to determine current dominant themes. We did not include country reports, meeting papers, or stakeholder correspondence in our review as these documents were not available to us. A.T.C. and K.B. independently reviewed the meeting reports for the following keywords and phrase combinations—Pacific OR Pacific Islands OR the names of each PICT AND universal health coverage OR UHC OR primary health coverage OR PHC. When identified, the section of the reports containing the keyword or phrase was reviewed and commentary specific to the PICTs was extracted. We define PICTs as the island nations and territories that fall within WHO’s Western Pacific Region (a list of PICTs is provided in Appendix A).

The second part of the study involved a review of the scientific literature published between 1 January 2015 and 31 July 2020 that discussed UHC in the PICTs. We systematically searched electronic databases Embase (Ovid interface; 1 January 2015 to 31 July 2020); MEDLINE(R) (Ovid interface; 1 January 2015 to 31 July 2020), and MEDLINE Epub Ahead of Print, In-Process & Other Non-Indexed Citations without Revisions (Ovid interface; 1 January 2015 to 31 July 2020) on 15 August 2020. The search included two clusters of terms: one related to ‘UHC’ and one to ‘PICTs’. Each cluster contained relevant medical subject heading (MeSH) and English and French language keyword combinations. Only articles available in English language were included. Our search strategy is presented at Appendix A. Articles identified were screened by two researchers (A.T.C. and K.B.) and included if they explicitly discussed UHC-related challenges, actions, or priorities in relation to a/the PICT/s. The reference lists of selected documents were reviewed to identify additional literature missed by the search.

From each eligible document (both grey literature (i.e., meeting reports) and peer-reviewed articles), the following was extracted. Data about (i) the document, including date of publication, title, full reference, author’s affiliation, and type of article (e.g., research report, lesson from the field or letter to the editor); and (ii) UHC challenges, responses, and priorities. These data were amalgamated in a tool developed in MS Excel^®^. We used a general inductive approach [15] to analyze these data. The generalized inductive approach applies a data coding and recoding system to identify recurring and dominant themes and condense raw textual data into a summarized format [15].

Literature was collected, data extracted, and analysis conducted between August and October 2020.

## 3. Results

We identified 17 documents including five Pacific Heads of Health meeting reports [16,17,18,19,20], three Pacific Health Ministers meeting reports [5,8,21], six WHO Western Pacific Regional Committee meetings reports [22,23,24,25,26,27], and three peer-reviewed papers [1,10,28]. Across these documents, five main themes emerged. These are presented below.

### 3.1. The Healthy Islands Vision Has (and Continues) to Have a Unifying Influence on Action for UHC

Evidence across the literature indicates that the Healthy Islands vision has provided a powerful call to action that has rallied decisionmakers’ attention in favor of health at the regional and national levels [1,3,29]. The vision’s resonance with Pacific decisionmakers is said to be because it frames health within an ecological worldview, an attitude that aligns with Pacific Islanders’ perception that health, environment, and culture are intimately linked [1]. Matheson et al. comment that the Healthy Islands vision encapsulates a sentiment of harmonization and unification that—in practical terms—aligns with a preference for collaborative action and settings approaches to address challenges, including those that impede health development [1].

While sentiment for the vision is strong, and initiatives are often framed as contributing to its overall goals, there is broad consensus that more could have been achieved over the last 20 years had greater attention been paid to implementation barriers, including the fragmentation of efforts driven by vertical programming and a focus on the development of central agencies’ capacity over that of provincial and local authorities who, in many PICTs, carry the responsibility for delivery of primary health care services [1,10]. Furthermore, while consistent reassertion of a shared vision may have encouraged solidary among PICTs, this sentiment has not always translated into operationalizable advice and action at local levels.

### 3.2. Adoption of Appropriate Service Delivery Models That Support Integrated Primary Health Care at the Community Level Is Needed

The delineation of packages of services, both preventive and curative, required to meet health care needs at the various levels of the health system, as well as having health care workers with the right skills mix in the right locations to deliver those services, are key strategies for UHC-aligned health sector reform across the PICTs [19]. First mentioned at the 2018 Pacific Heads of Health meeting, these themes appear linked to the release of the report Universal health coverage on the journey towards Healthy Islands in the Pacific [9] the previous year. The report (and subsequent dialogue) encourages PICT decisionmakers to consider models by which health services are delivered that integrate community and primary health care (PHC); increasing the share of resources allocated to lower level health facilities and community-based services for delivery of PHC; and improving the managerial, administration, or supervisory capacity of the health system to ensure that resources reach the periphery of the health system and are well used [9,10].

Strong and accessible primary health care (i.e., the first contact a person has with the health system (typically a general practitioner or community health worker) [30] is deemed essential for the achievement of UHC and is directly supportive of the Healthy Islands vision [18,20,25] In 2017, a report from the 5th Pacific Heads of Health Meeting noted that PHC “has been neglected for years in the region and yet increasingly is considered an important starting point for UHC” [18]. Calls for the ‘revitalization’ of PHC were repeated in 2019 at the 7th Pacific Heads of Health Meeting, where participants “acknowledged the importance of strengthening their PHC systems and the need for political will for health sector reform.” [20] Pacific Health Ministers reiterated this in the same year through their commitment to “strengthening PHC as the key delivery strategy for UHC in the Pacific” [21]. This is in line with the international commitment made at the Global Conference on PHC in 2018 and captured in the conference’s declaration, the Declaration of Astana (2018) [31].

### 3.3. Human Resources for Health Are Critical If Efforts to Achieve UHC Are to Be Successful

Inadequate human resources for health have been a constant theme of senior Pacific health leader meetings, with opinions raised at these meetings reflected in the literature. There is recognition given to the importance of increasing the number of skilled health workers to achieve UHC, with some progress noted in the number of doctors, but less in scaling up additional frontline workers. The health workforce profile in PICTs shows that, while many have reached the critical threshold required to meet the SDGs, some PICTs with bigger populations (i.e., PNG, Vanuatu, the Solomon Islands, and Samoa) have not [29]. In PNG, for instance, it is estimated that 3.5 million people do not have access to a doctor within their district and that the country will need 17,600 additional skilled health workers to meet the SDG goal [32].

Delegates at the 66th Regional Committee Meeting for the Western Pacific [22] noted that the dominant SDG indicator by which workforce sufficiency is measured (i.e., the number of skilled health workers per 10,000 population) does not take into account the challenges in delivering care to highly decentralized populations dispersed over thousands of islands and atolls. While Pacific-specific thresholds to guide workforce adequacy have not been developed, individual PICTs have devised human resource for health indicators and targets that take into consideration local conditions [33,34].

Heads of Health and Ministers’ meetings note that domestic capacity to train adequate numbers of health workers is lacking in many of the smaller PICTs and that there is significant variation in education and training available to health practitioners. Pacific Health Ministers noted in 2017 that more than 250 health-focused courses are on offer across the region with “varying levels of curriculum standards, academic support, [and] education and teaching materials” [8]. This has raised concerns about both the quality of education and training offerings and their relevance to the needs of Pacific populations [8,19].

Lack of domestic training capacity has required either the ‘importation’ of staff from overseas or the sending of nationals for training at education institutions overseas. Both options are expensive and logistically challenging. Initiatives to train Pacific Islander doctors abroad, notably in Cuba [28,35,36], have helped address human resources for health gaps and provide much-needed capacity in health. For example, the number of doctors in Kiribati increased from 18 to 51, with 23 of the new doctors Cuban trained. In the Solomon Islands, the number of doctors in the same period grew from 79 to 170, with the return of 74 medical graduates from Cuba between 2014–2018, and another 34 expected to graduate by 2025. Cuban trained doctors also contributed to significant increases in the number of doctors in Tonga, Tuvalu, Vanuatu, and Nauru [28,37]. However, these initiatives are not without problems with delegates to the 2016 Heads of Health meeting raising concerns for the ‘job readiness’ of returning overseas educated doctors and the resource-intensive need to provide additional primary care-focused training and mentorship [10,17].

Ministers note that a response to the health workforce deficit will require a nuanced approach considering the different islands’ contexts. The challenge will be to train and/or recruit adequately skilled staff, fund and support their deployment to rural areas where there is typically the greatest need and support them once there. At the 13th Pacific Health Ministers Meeting in 2019, this was acknowledged in the commitment to “develop training programs targeted at isolated medical practitioners and [implement] effective health workforce retention strategies” [21].

Ministers highlight that global disparities and shortages in the supply of healthcare workers exist, with high to severe shortages reported in many PICTs, where the need for health workers is most significant. Central to their concerns is the emigration of health workers to other PICTs and Pacific-rim countries and the deficits in capacity left. Research shows that this “continuing exodus of skilled health care professionals” [5] is high, and Australia and New Zealand’s contribution to this ‘brain drain’ has increased from 455 Pacific-born doctors and 1158 Pacific-born nurses and midwives reported as working in Australia in 2006, to 607 doctors and 2954 nurses in 2016 [38]. The result is increased deficits in human resources for health in these contexts, weaker health systems and the undermining of regional capacity development efforts [38].

### 3.4. Access to Reliable Health Information Is Central to Health Sector Improvement

A reliance on antiquated paper-based health information reporting and the lack of data quality standards/checks are noted challenges to the reliable monitoring and reporting of PICT populations’ health status and the availability of evidence to base system development decisions in support of UHC [29].

The limitations of PICTs’ health information systems are eloquently articulated in the foreword to the second Healthy Islands Monitoring Framework [HIMF] progress report when the author writes, “Many datasets [on which accurate monitoring relies] are works in progress that require improvement, including capacity-building and overall strengthening of health information systems” and, “[the] primary challenge is to ensure that monitoring systems are strengthened or introduced to ensure data are collected regularly and with a high degree of accuracy” [29]. Underpinning these efforts is the recognition that investment is required to strengthen national health information systems [29]. Delegates have noted that these efforts should include improved data sources; increased capacity to manage and analyze data and ensure information is translated into action; and work to “reinforce equity-oriented health information systems with more disaggregated data across age, sex, geography, household income levels and other characteristics appropriate to the country context” [8].

Pacific health leaders recognize the limitations and highlight the transformative opportunities prudent digitization of health information offers health systems—both in terms of the opportunity to enhance the continuum of care provided and for macroscopic health system performance oversight and reporting [19,20,26].

### 3.5. While Not a Panacea for All Challenges, Digital Health Offers Many Opportunities

Adoption of digital health interventions, through a range of applications such as eLearning, telemedicine, and digital health information systems, has been raised by delegates to senior health fora as an opportunity to support efforts to address UHC. For example, attendees at 2019 Heads of Health meeting highlight telemedicine as a tool that may help overcome challenges in the delivery of health services in rural and remote areas [20].

Delegates to the 2020 WHO Regional Committee Meeting noted that there has been a rapid acceleration in the use of digital health in some countries while others remain in the early stages of adoption; and that this has resulted in a ‘digital divide’ between countries. They highlighted the UHC principles of equity and stress the need to explore ways and means by which well-advanced countries can support those with fewer resources to capitalize on the opportunities digital health may offer [27].

Discussion at the 2018 Regional Committee Meeting noted the “roll-out of information and communications technology for health service delivery was uneven across the Region” [25] but that the recently developed Regional Action Agenda on Harnessing E-Health for Improved Service Delivery in the Western Pacific [39] was a tool to guide countries’ adoption of e-health. This document sought to guide the rationale use of e-health through cost-effective means with an emphasis on the “development of appropriate infrastructure, information-sharing, and privacy and security mechanisms” [25]. Strategies to ensure the development of effective digital health interventions featured again at the 2019 7th Pacific Heads of Health Meeting, where delegates from Vanuatu stressed the need for investment in infrastructure to support digital health roll-out in the Pacific together with legislation to guide information-sharing and data privacy. Furthermore, delegates stressed the need to include end-users (i.e., health staff) in developing digital health strategies to garner buy-in and support for e-health initiatives and ultimately increase the likelihood of sustained utilization of digital tools [20]. The importance of ongoing external technical assistance to support PICTs’ transition to digital platforms while the human resource capacity for domestically managed e-health is developed, Pacific regional cooperation for the sharing of insights and development costs, and the need for visionary leadership to drive the adoption of digital health were highlighted [20].

## 4. Discussion

This study’s thematic analysis of 17 grey and scientific peer-reviewed literature from 2015 to 2020 on topics related to UHC priorities for Pacific islands health authorities yielded five key themes. The five themes were (i) the Healthy Islands vision has (and continues to have) a unifying influence on action for UHC; (ii) adoption of appropriate service delivery models that support integrated primary health care at the community level are needed; (iii) human resources for health are critical if efforts to achieve UHC are to be successful; (iv) access to reliable health information is core to health sector improvement; and, (v) while not a panacea for all challenges, digital health offers many opportunities.

The enormity and complexity of the challenges Pacific Island decisionmakers face in delivering care to their populations cannot be understated. Small and isolated populations, chronic workforce limitations, weak governance arrangements, ageing and inadequate quality health facilities, limited investment in information and communication technologies, supply chain and logistics difficulties, and fragile health systems among other issues conspire to challenge the delivery of PHC across PICTs. These highly contextual challenges, together with the significant disrupting effect the coronavirus pandemic has had on capacity to implement health development programs across the Pacific and a likely shift in models by which technical assistance will be delivered in a post-pandemic world, mean that a closer and more streamlined alignment between national priorities, domestic capacity to implement programs to achieve these priorities, and international investment in health sector development will be needed.

The continued discussion about UHC demonstrates the importance senior health leaders have placed on it and health system strengthening approaches more broadly. The repeated reassertion of a shared vision around which Pacific Island leaders can rally has been unifying; however, challenges to UHC are often highly contextual and require tailored local solutions. Where success in regional approaches has been seen, such as in health information, considerable investment has been made at national and subnational levels to interpret and implement inherently generic regional policy and programmatic advice. Looking ahead, the utility of the Healthy Islands vision may be enhanced if it is able to differentiate the UHC-related challenges for which regional responses are warranted, and articulate what, where, and how the national and/or subnational actions that are required to realize the vision can be achieved. This would support the alignment of and investment in action required across the spectrum (i.e., from regional to local) to see results.

We found evidence of strong support for the Healthy Islands vision as a tool that has focused action and garnered support for UHC from both traditional and non-traditional stakeholders. However, to realize this vision, a realistic understanding of the political, human resource, and economic investments needed is required. This is particularly relevant for PICTs where rapid population increases, growing demand for specialized services, the need to provide care to a rapidly urbanizing population while maintaining services for rural and remote communities, and scaling health care in response to a growing non-communicable disease epidemic pose significant challenges.

Both financial and human resources required for health system reform are—for most PICTs—above what is domestically available. The significant disruptive effect COVID-19 has had on service delivery, population health, and—more broadly—capacity to raise budget allocations for health, together with the uncertainty it brings for implementation of the medium- to long-term health development agenda raises concern that progress made may stagnate or retreat [40]. Hence, the sustained support of donors such as Australia, New Zealand, the United States, Japan, and China and development partners including WHO and the Secretariat of the Pacific Community (SPC) will be required for the foreseeable future. Determining the most helpful contributions external agencies can make will inevitably be context-specific; however, they should align with an ethos of health systems strengthening and promoting health equity, national leadership, and sustainability. The global and regional action frameworks for UHC [30,41,42], together with PICTs’ National Health Strategic Development Plans provide endorsed, established, and widely used guidance for UHC-aligned health sector investment.

Interestingly, despite the call for increased financing for PHC in the 2017 WHO report, Universal health coverage on the journey towards Healthy Islands in the Pacific [11], we found little evidence of the issue being discussed at senior level meetings. This may be because there are relatively high government and donor allocations for health and relatively low out of pocket costs, perhaps lessening the urgency of financing strategies directly linked to UHC and PHC, or because such matters are discussed in different forums. With the achievement of health service delivery goals so resource-sensitive in PICTs, opportunities to explore, compare, analyze, and document the success and failure of different approaches to PHC delivery and financing are needed. Consideration of existing financing analysis, including those developed by the World Bank for some PICT [43,44,45], would seem to warrant discussion at senior health fora. Furthermore, the application of implementation science-based research to build evidence for what constituted ‘good’ PHC/UHC policy and practice in the Pacific Island context is required.

We note that Regional Committee Meetings included a range of technical agenda items relevant, if not central, to UHC that PICT delegates would likely have commented on but for which there is no evidence in the meeting reports. These include items relating to the regional action frameworks and agendas for UHC [42], on regulatory strengthening, convergence and cooperation for medicines and the health workforce [46], on improving hospital planning and management [47], and on strengthening legal frameworks for health in the SDGs [48]. While not discussed in this paper, these broader issues and the related focus on access, quality, equity, and financial protection are central to the achievement of UHC in the PICTs and must not be overlooked. Health authorities and their development partners will need to consider a more comprehensive suite of setting-specific evidence (including internal reports and data) when assessing, planning, and implementing activities in response to setting-specific UHC challenges. While there is no ‘one-size-fits-all’ plan for achieving UHC, PICT health authorities may look to the Western Pacific Region’s Action Framework for UHC and its 15 action domains to frame a comprehensive whole-of-system approach to health sector strengthening [42]. The 15 action domains relate to five core attributes of a high-performing health system; these are quality, efficiency, equity, accountability, and sustainability and resilience [42] (Table 1).

Our review found growing attention to the opportunities digital technologies offer UHC in the Pacific. Specific mention of the value of digitization of health records, telemedicine, online information sharing, and e-learning may offer PICT health authorities’ efforts to improve health informatics (and hence capacity to monitor health system performance), provide remote health care to rural and remote communities, and for health worker training and support were noted. The COVID-19 pandemic has highlighted the opportunity digital health offers health systems with many countries accelerating the roll-out of technology-enabled applications—including telemedicine, direct and indirect client communication, and e-learning—to overcome access and delivery issues that have arisen. Recent infrastructure improvements across the Pacific, including the opportunity for high-speed internet due to the laying of submarine fiberoptic cables, and increasing internet connectivity (primarily due to increasing smartphone ownership) have created opportunities to integrate and capitalize on digital health for the delivery of PHC in the islands. Early examples of digital health adoption in the Pacific Islands include the development of an electronic health information system, and the gradual transition from paper to electronic records in the Solomon Islands [49], use of internet-connected tablets for data collection, information sharing, and clinical decision support in PNG [50], and installation of very small aperture terminals that allow health facilities across the islands of Tuvalu to be ‘digitally linked’ through satellite communication (personal communication, K. Borget, 18 November 2021). These examples, as well as an increased attention to building information technology and data analysis skills through initiatives such as the Strengthening Health Interventions in the Pacific (SHIP) initiative and the field epidemiology training programs, showcase the opportunity that prudent adoption of digital technology for health service delivery offers efforts to achieve UHC in resource constrained and challenging contexts.

The Public Health Information Network (PHIN) provides direction for adopting digital technologies for health in the Pacific. While advocates for digital health, the PHIN note that the adoption of technology is complex and should not be seen as a panacea for UHC challenges. Health authorities must weigh up the costs in developing and maintaining digital health applications with the long-term savings and improvements on offer. To ensure value is achieved, the process of digital adoption should be framed as an integral part of health sector reform and not a series of discrete program-specific interventions. This has been demonstrated by the staged adoption of digital health information systems in many PICTs and integration of mHealth approaches to specific surveillance programs, such as the Pacific Syndromic Surveillance System.

Our review is not without limitation. Most notably, the review’s scope was limited to analyzing the final reports from senior health meeting fora and peer-reviewed literature in which UHC was discussed. Furthermore, our data extraction process relied on there being a combination of terms related to UHC and the Pacific islands which—while broad—likely missed discussion relevant to UHC but not specifically identified as such (e.g., leadership or health sector reform) or for which PICT delegate contributions was not recorded. While our search strategy included the French language names of the three French PICTs, we excluded articles not available in English and, therefore, may have overlooked some literature. Our data were drawn from regional sources which potentially missed setting-specific differences in the barriers to UHC achievement, the action being taken in response to local challenges, and opportunities available. Furthermore, the literature tended to focus on challenges and not solutions, and hence discussion about the policy and program responses may have been missed. Nevertheless, to the best of our knowledge, this review provides the first analysis to synthesize the high-level UHC-related discussions and priorities of PICT policymakers. This is important as it provides evidence to support the alignment of domestic and international investments in health sector development in the Pacific Islands with an indication of the current priorities.

## 5. Conclusions

This paper analyzed documents from regional health leader fora and the literature to provide insights into the UHC-related priorities of PICTs. We found evidence that the Healthy Islands vision has been a tool to garner support for UHC but note that to realize the vision, a realistic understanding of necessary political, human resource, and economic investments is required. The significant disruptive effect of COVID-19 and the uncertainty it brings for health sector development in the medium- to long-term raises concern that progress made in recent years may stagnate or retreat. While a one-size-fits-all approach to address UHC is not appropriate across the Pacific islands, there are well developed and agreed development frameworks around which PICTs – as individual States and as a collective block – and their development partners may rally.

## Figures and Tables

**Table 1 ijerph-19-04108-t001:** Health system attributes and action domains for universal health coverage.

Health System Attributes	Action Domains for UHC
Quality	Regulations and regulatory environment
	Effective, responsive individual and population-based services
	Individual, family, and community engagement
Efficiency	Health system architecture to meet population needs
	Incentives for appropriate provision and use of services
	Managerial efficiency and effectiveness
Equity	Financial protection
	Service coverage and access
	Non-discrimination
	Government leadership and rule of law for health
	Partnerships for public policy
	Transparent monitoring and evaluation
Sustainability and resilience	Public health preparedness
	Community capacity
	Health system adaptability and sustainability

Adapted from World Health Organization. Universal Health Coverage: Moving Towards Better Health: Action Framework for the Western Pacific Region, Manila, Philippines, 2016 [42]. Creative commons license: CC BY-NC-SA 3.0 IGO.

## Data Availability

Not applicable.

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
