# Peer review of "Universal Health Coverage and the Pacific Islands: An Overview of Senior Leaders’ Discussions, Challenges, Priorities and Solutions, 2015–2020"

_ijerph, 2022, doi:10.3390/ijerph19074108_

Round 1

Reviewer 1 Report

As a suggestion I would remove brackets in your sentences or words used. 

This is a well written paper with some key findings. 

Reference 35: Werle. C.  is listed down twice. Please remove one of them.       

Author Response

We thank reviewer 1 for their comments. Please find responses to each point raised below.

Comment 1. As a suggestion I would remove brackets in your sentences or words used. 

-->We have reviewed the manuscript and removed use of brackets, where able.

Comment 2. Reference 35: Werle. C.  is listed down twice. Please remove one of them.

--> We have removed the repeated words in reference 35.

Reviewer 2 Report

 Review reports: Universal health coverage and the Pacific Islands: an overview of senior leaders’ discussions, challenges, priorities and solutions, 2015- 2020

A brief summary (one short paragraph) outlining the aim of the paper, its main contributions and strengths.

The article was clear and relevant with recent articles included in this review – which included articles in the previous 5 years. While the purpose of the review is not clear, they did mention they were analysing discussions, resolutions and recommendations to align to domestic and international investment, the article maximised the challenges leaving very little information on the resolutions. At some point, it gave conflicting information whilst the discussion section did highlight some of the resolutions.

The generalising statements across PICTs can be disputed as some may not have all the challenges, some are improving and the fact COVID halted development may have been an obstacle to achieving their goals.

The paper is timely, it can assist policymakers and health promotion officials to track their development. It is an appropriate article but would just need to make room to highlight some aspirations that could have been achieved or targets that can be met if they implement certain strategies.

General concept comments

The article is an interesting one that draws the on the discussions of heads of health meetings and could have been a very interesting paper. While the aim of the paper was to promote alignment of domestic and international investments in health sector development, the results seem to highlight the weaknesses or the challenges and do not stress the responses or recommendations. I feel that to have a balanced view as proposed, they could have aligned these findings focussing on recommendations or resolutions and compared to international standards.

In the discussion about the quality of training being questioned at the Heads of Health meetings while at the same breath lamenting the “brain drain” which is an indication of the recognition of programmes or curriculum content.

While the methodology did state that the use of general inductive approach, the analysis focussed on the challenges at most and if there were resolutions or recommendations proposed, it could have balanced the finding and to align to international investment would highlight the most important areas for the “right” UHC investment. For eg., Section 3.4 did highlight access to reliable HIS about regular and accurate data collection. In recent meetings WHO and SPC have highlighted the training of HIS team in PICTs through the SHIP project nor called Field Epi which is a response and a resolution to the ICT challenges. Its relation to UHS means timely accurate dissemination of health status.

Review: commenting on the completeness of the review topic covered, the relevance of the review topic, the gap in knowledge identified, the appropriateness of references, etc.

I commend the authors for writing up the article – it is timely and could be a useful article for those in the health promotion field or even in health policy, a baseline for future articles in assessing these regional meetings summaries which is useful in identifying and enabling policymakers to achieve their targets of what has been discussed. I just feel that it could reduce bias if I also highlighted some positives, although appearing vaguely in the discussion, it could be well placed and appropriate in results and maybe more elaborate discussion.

I believe these types of articles is to be welcomed and it will be useful to policymakers. It does address pertinent issues but discussions must be cautioned that it is not biased or one-sided. Generalising findings across all PICTs may be a challenge given their varying economic status and national health standards therefore some findings can be challenged if it is generalised.

Specific comments

-Line 179-180 of page 4 talks about the (questionable) quality of education and training in the Pacific however line 205 talks about “brain drain” to Australia and NZ which means they recognize the quality of training received in the Pacific. While it questions the quality of curriculum – it also highlights the strength in the same tone which could confuse readers. Maybe a suggestion is to:

-> Identify institutions or programmes that are questionable (if data is available)

OR

->Acknowledge the brain drain as recognition of the quality of training and/curriculum offered in the Pacific

-Page 5 – 3.4 on ICT information

->Strongly agree on the improvement of ICT in Pacific as previous meetings have also highlighted existence of “data cemetery” in the Pacific with quality being questioned on accuracy and reliability.

->Programmes being offered by SPC for Pacific Islands states in training health personnel on data management e.g. SHIP or Field EPI programme which I think was presented in the 2017 and 2019 meetings.

-Line 251 page 5, rational or rationale – check spelling

-Page 8, paragraph 1 does talk about PHIN, it would also benefit the readers to include the setting up of early warning systems as a strength of digital technology to PICT’s. This system enables remote settings to be informed of prevalent infections that could affect out-lying islands.

Author Response

We thank reviewer 2 for their comments. Please find responses to each comment below.

General concept comments

Comment 1. The article is an interesting one that draws on the discussions of heads of health meetings and could have been a very interesting paper. While the aim of the paper was to promote alignment of domestic and international investments in health sector development, the results seem to highlight the weaknesses or the challenges and do not stress the responses or recommendations. I feel that to have a balanced view as proposed, they could have aligned these findings focussing on recommendations or resolutions and compared to international standards.

--> We acknowledge the reviewer’s comment about the balance between identification of the challenges to achieving UHC and responses, but note that our presentation of results and the subsequent discussion are based on the data collected which, perhaps due to the nature of meeting records drawn on, were focused on challenges and not solutions. In response to the comment, we have expanded the limitations paragraph of the paper (at line 419) to include, “Further, the literature tended to focus on challenges and not solution and hence discussion about the policy and program responses may have been missed.”

We note that the research was not designed to evaluate results against international standards, rather the paper aimed to document key discussions and resolutions of Pacific leaders in order to document and generate insight into leaders’ priorities. To temper the discussion and address the reviewer’s comment, we have amended text in the to highlight responses to UHC challenges faced. For example, “Recent infrastructure improvements across the Pacific, including the opportunity for high-speed internet due to the laying of submarine fiberoptic cables, and increasing internet connectivity (primarily due to increasing smartphone ownership) have created opportunities to integrate and capitalize on digital health for the delivery of PHC in the islands. Early examples of digital health adoption in the Pacific Islands include the development of an electronic health information system, and the gradual transition from paper to electronic records in the Solomon Islands [49], use of internet-connected tablets for data collection, information sharing, and clinical decision support in PNG [50], and installation of very small aperture terminals that allow health facilities across the islands of Tuvalu to be ‘digitally linked’ through satellite communication (personal communication, K. Borget, 18 November 2021). These examples, as well as an increased attention to building information technology and data analysis skills through initiatives such as the Strengthening Health Interventions in the Pacific (SHIP) initiative and the field epidemiology training programs, showcase the opportunity that prudent adoption of digital technology for health service delivery offers efforts to achieve UHC in resource constrained and challenging contexts.”

Comment 2. In the discussion about the quality of training being questioned at the Heads of Health meetings while at the same breath lamenting the “brain drain” which is an indication of the recognition of programmes or curriculum content.

--> We note that the two issues (senior leader’s comments about the quality of training opportunities and ‘brain drain’, while presented in the same section of the paper, are discussed as different theme points and in different paragraphs. We have reviewed the manuscript and do not feel the presentation is confusing or contradictory. We note that drivers of ‘brain drain’ are complex and beyond the scope of this review-based research paper.

Comment 3.  While the methodology did state that the use of general inductive approach, the analysis focussed on the challenges at most and if there were resolutions or recommendations proposed, it could have balanced the finding and to align to international investment would highlight the most important areas for the “right” UHC investment. For eg., Section 3.4 did highlight access to reliable HIS about regular and accurate data collection. In recent meetings WHO and SPC have highlighted the training of HIS team in PICTs through the SHIP project nor called Field Epi which is a response and a resolution to the ICT challenges. Its relation to UHS means timely accurate dissemination of health status. 

--> The comment appears to relate to 2.1 and hence the response above somewhat addresses the points made. In addition, we have added text to the discussion section starting at around line 354 that summarises the recommended health systems areas on which countries should focus to progress UHC. The newly added text reads, “The 15 action domains relate to five core attributes of a high-performing health system; these are quality, efficiency, equity, accountability, and sustainability and resilience [42] (Figure 1).

Figure 1. Health system attributes and action domains for universal health coverage.

Health system attributes

Action domains for UHC

Quality

Regulations and regulatory environment

Effective, responsive individual and population-based services

Individual, family, and community engagement

Efficiency

Health system architecture to meet population needs

Incentives for appropriate provision and use of services

Managerial efficiency and effectiveness

Equity

Financial protection

Service coverage and access

Non-discrimination

Government leadership and rule of law for health

Partnerships for public policy

Transparent monitoring and evaluation

Sustainability and resilience

Public health preparedness

Community capacity

Health system adaptability and sustainability

Adapted from [42].

Comment 4. I commend the authors for writing up the article – it is timely and could be a useful article for those in the health promotion field or even in health policy, a baseline for future articles in assessing these regional meetings summaries which is useful in identifying and enabling policymakers to achieve their targets of what has been discussed. I just feel that it could reduce bias if I also highlighted some positives, although appearing vaguely in the discussion, it could be well placed and appropriate in results and maybe more elaborate discussion.

I believe these types of articles is to be welcomed and it will be useful to policymakers. It does address pertinent issues but discussions must be cautioned that it is not biased or one-sided. Generalising findings across all PICTs may be a challenge given their varying economic status and national health standards therefore some findings can be challenged if it is generalised.

--> We agree with the reviewer wholeheartedly but reiterate that the scope of our paper and the method applied to rely on data collected from senior health meetings and the literature. As noted above, we have included examples of good practice in response to UHC challenges in the discussion section and feel the paper reports results in a clear and neutral way. With regards to generalisability, at line 359 our paper notes that each PICT is unique and faces a different set of challenges and that a one size fits all approach to response is not appropriate. In addition, we have added text to the limitation paragraph that reiterates that caution should be applied when generalising the findings of this research. The passage added reads, “Further, our data was drawn from regional sources which potentially missed setting-specific variances in the barriers to UHC achievement, the action being taken in response to local challenges, and opportunities available.”

 Specific comments

Comment 5. -Line 179-180 of page 4 talks about the (questionable) quality of education and training in the Pacific however line 205 talks about “brain drain” to Australia and NZ which means they recognize the quality of training received in the Pacific. While it questions the quality of curriculum – it also highlights the strength in the same tone which could confuse readers. Maybe a suggestion is to: identify institutions or programmes that are questionable (if data is available) OR acknowledge the brain drain as recognition of the quality of training and/curriculum offered in the Pacific

--> We thank the reviewer for their practical suggestions but feel that the results of our research are not sufficient to make such judgement. Further, we note that the issue of ‘brain drain’ is complex and warrants in depth investigation to generate insight to inform policy and practice. As noted above, given the discussion about perceived quality of training and ‘brain drain’ are presented in separate paragraphs with no direct link we feel the presented text is clear.

 Comment 6.  -Page 5 – 3.4 on ICT information

a. Strongly agree on the improvement of ICT in Pacific as previous meetings have also highlighted existence of “data cemetery” in the Pacific with quality being questioned on accuracy and reliability.

--> We acknowledge the reviewer’s agreement.

b. Programmes being offered by SPC for Pacific Islands states in training health personnel on data management e.g. SHIP or Field EPI programme which I think was presented in the 2017 and 2019 meetings.

--> In response, content has been added at around line 383 highlighting the value of the SHIP and Field epidemiology training initiatives. It reads, “These examples, as well as increased attention to addressing skills development needs through initiatives such as the Strengthening Health Interventions in the Pacific (SHIP) and field epidemiology training programs, showcase the opportunity that prudent adoption of digital technology for health service delivery offers efforts to achieve UHC in resource constrained and challenging contexts.”

Comment 7. -Line 251 page 5, rational or rationale – check spelling

--> The error has been addressed.

Comment 8. -Page 8, paragraph 1 does talk about PHIN, it would also benefit the readers to include the setting up of early warning systems as a strength of digital technology to PICT’s. This system enables remote settings to be informed of prevalent infections that could affect out-lying islands.

--> In response, text has been added around line 383 that reads, “These examples, as well as an increased attention to building information technology and data analysis skills through initiatives such as the Strengthening Health Interventions in the Pacific (SHIP) initiative and the field epidemiology training programs, showcase the opportunity that prudent adoption of digital technology for health service delivery offers efforts to achieve UHC in resource-constrained and challenging contexts.”

Reviewer 3 Report

page 7, line 321-325.  "We found little evidence of the [financing] issue being discussed at senior level meetings".  "May be because there are relatively high government and donor allocations for health and relatively low out-of-pocket costs".   

These findings interested the reviewer much because financing is the biggest challenges to the countries already with UHC.     

Also it is a nice surprize for the reviewer that PICT leaders seem not to be bothered by socio-economic issues such as poverty, inequality and financial protection from disease, all of which UHC is expected to remedy the issues.   

The reviewer expects the authors to elaborate more in depth in these matters.    

Author Response

We thank reviewer 3 for their comments. Please find responses to each comment below.

Comment 1. page 7, line 321-325.  "We found little evidence of the [financing] issue being discussed at senior level meetings".  "May be because there are relatively high government and donor allocations for health and relatively low out-of-pocket costs".  These findings interested the reviewer much because financing is the biggest challenges to the countries already with UHC.     

--> We thank the reviewer for sharing that this aspect of our work was of interest.

Comment 2. Also it is a nice surprize for the reviewer that PICT leaders seem not to be bothered by socio-economic issues such as poverty, inequality and financial protection from disease, all of which UHC is expected to remedy the issues.  The reviewer expects the authors to elaborate more in depth in these matters.

--> In response, additional text has been added around line 354 that reads, “While not discussed in this paper, these broader issues and the related focus on access, quality, equity, and financial protection are central to the achievement of UHC in the PICTs and must not be overlooked.”